# Self-care needs among international migrants and travellers: A systematic review and meta-synthesis

Antonius Nugraha Widhi Pratama[1,2]*, Brahmaputra Marjadi[3],
Jack Charles Collins[1], Rebekah Jane Moles[1], Carl Richard Schneider[1]

1 The University of Sydney School of Pharmacy, Camperdown, New South Wales, Australia, 2 Faculty of Pharmacy, University of Jember, East Java, Indonesia, 3 School of Medicine, Western Sydney University, Campbelltown, New South Wales, Australia

☯ These authors contributed equally to this work.
* antonius.pratama@sydney.edu.au

## Abstract

International travellers and migrants perform self-care to maintain their health in their destination countries. The aim of this review was to determine the self-care needs of international travellers and temporary migrants, and to assess how these needs align with existing self-care frameworks. We conducted a systematic review and meta-synthesis, searching MEDLINE, Embase, International Pharmaceutical Abstracts, PsycINFO, and CINAHL for studies on self-care among international travellers or migrants without limiting date and language. Two reviewers performed title and abstract screening after deduplication, followed by full-text screening. Two reviewers used the Center for Evidence-Based Medicine's (CEBM) Critical Appraisal of Qualitative Studies to assess the quality of selected articles. Discrepancies were resolved by consensus with a third reviewer. Thomas and Harden's thematic synthesis was applied to synthesise the included literature. El-Osta's Self-Care Matrix (SCM) was used as a sensitising concept and to map conceptual similarities of descriptive themes. The protocol was registered in PROSPERO (CRD42022372693). The searches retrieved 2,394 articles, of which 17 were considered acceptable quality and included in the meta-synthesis, totalling 769 participants. Two studies were conducted among travellers and 15 among migrants. Twenty-one descriptive themes were generated, and five analytical themes were synthesised. All descriptive themes, except social engagement, were mapped to the SCM, with partial convergences occurring for self-care products and medicines, self-treatment, and treatment adherence. This meta-synthesis identified five self-care needs: self-care empowerment, mutual understanding, healthcare challenges and opportunities, preventive self-care, and self-care facilitation. Travellers' and migrants' reduced social capital (resource-gaining from social networks) may impact their ability to self-care, and social engagement is a form of social capital important for their self-care.

**Data availability statement:** All relevant data are within the manuscript and its Supporting Information files. Our meta-synthesis used results or findings of included studies as data. The included studies were published and available in the literature.

**Funding:** ANWP receives a PhD scholarship from the Indonesian Endowment Fund for Education (LPDP) to study a PhD at The University of Sydney School of Pharmacy, New South Wales, Australia. The scholarship number is 202201223207898. The sponsor does not have any role in the study design, data collection and analysis, decision to publish, or the preparation of the manuscript.

**Competing interests:** The authors have declared that no competing interests exist.

## Introduction

International travel and migration are driven by complex factors such as gaining new experiences and knowledge, and better livelihood and security [1]. Post-COVID-19 pandemic, international travel is expected to rise. More efficient and accessible air travel and communication technology advancements are among the facilitators of this mobility [1]. International tourist arrivals recorded in 2024 were 1,445 million globally, slightly below 2019 [2]. Approximately 281 million people were migrants in 2020, which increased from 173 million in 2000 [3]. Migrant workers constituted 69% (or 169 million) of international migrants in 2019 [4].

International travel and migration are different but related [5]. The identity boundary between international tourists and migrants is not lucid as actors mix mobility purposes, for example, "working holiday visa" holders and "digital nomads", referring to travellers who also work remotely from place to place [6]. This blurred boundary is often found between international travellers and migrants whose stay in the destination country is relatively short or transient compared to long-term and permanent migrants.

Regardless of the primary purpose of mobility, migration and transience disrupt social capital (the aggregate of resources embedded in social networks) and influence self-care practices [7,8]. Reduced social capital may limit access to health information and services in the destination, making social engagement important for rebuilding these networks. Moreover, international travellers and migrants need to understand and adapt to the health system in their destination. Tourists, characterised by a short-term visit, may not need access to healthcare as intensively as long-term migrants, except in emergencies [9]. Familiarity with and availability of support in destination countries can help ease barriers to accessing healthcare, but not all international travellers and migrants have these luxuries. Barriers to and facilitators of healthcare access and utilisation in the destination countries were identified among migrants and international students, but are less discussed among tourists [10,11]. However, it remains unclear if barriers to accessing healthcare may or may not be a driver for self-care.

Self-care differs from related constructs, such as self-management (focused on managing chronic diseases), self-efficacy (confidence in performing tasks), and empowerment (capacity building) [12,13]. This review adopted WHO's definition of self-care as "the ability of individuals, families and communities to promote health, prevent disease, and maintain health and to cope with illness and disability with or without the support of a health-care provider" [14]. We define self-care needs as conditions and resources that enable individuals and communities to engage effectively in self-care. The scope of self-care includes health promotion, prevention and control of disease, self-medication, provision of care to dependent people, seeking health services when necessary, and rehabilitation [14]. Individuals perform self-care, including giving self-care to their children or a neighbour. Pre-departure preparations, such as travel vaccinations and insurance, can be included in self-care [1,9]. Awareness of the risks of misadventures is also self-care. Migrant workers need to understand the health and safety aspects of their tasks to prevent occupational accidents.

Minor health-related events often prompt travellers and migrants to self-medicate or seek care [9,15]. Upon returning to their country of origin, international travellers must know they may import infectious diseases [9].

The breadth of the scope of self-care has led to several self-care theories, models, and frameworks [12,13,16]. However, to our knowledge, no self-care theories, models, or frameworks have been specifically applied to travel and migrant populations to understand their self-care needs. The identification of self-care needs would assist in developing strategies for safe and effective self-care practices for international travellers and migrants. Therefore, we conducted a systematic review and meta-synthesis of qualitative literature to identify self-care needs among international travellers and temporary migrants, and to examine how these needs align with the concept of self-care.

## Methods

This meta-synthesis was part of a larger study to identify self-care needs among travellers, differences between travellers based on self-care prioritisation, and measures to meet these needs. The review protocol was registered in PROSPERO (CRD42022372693), and findings were reported according to the Enhancing Transparency in Reporting the Synthesis of Qualitative Research (ENTREQ) (S1 Appendix) [17]. The systematic reviews followed the Preferred Reporting Items for Systematic reviews and Meta-Analyses (PRISMA) checklist (S2 Appendix) [18]. Ethics approval was not applicable as this meta-synthesis analysed published data.

### Scope, inclusion, and exclusion criteria

This review included studies on international travellers and migrants with voluntary and transient mobility. We included international tourists, backpackers, international students, business travellers and transient or seasonal migrant workers. Travellers and temporary migrants were analysed as a single conceptual category because they share similar adaptation challenges in unfamiliar environments, such as limited social capital, social positions, language difficulties, cultural differences, and barriers to healthcare access [19–21]. The synthesis was to identify and highlight commonalities across these populations rather than differences. Permanent migrants were excluded due to their long-term integration into destination health systems, which differs substantially from the temporary context of transient mobility [21]. Refugees and asylum seekers were also excluded because their health needs may be shaped by forced displacement and trauma, requiring distinct analytical frameworks [19].

A systematic search was conducted in MEDLINE, Embase, International Pharmaceutical Abstracts, PsycINFO, and CINAHL databases to find qualitative, multi-methods, and mixed-methods primary research articles focusing on self-care and travel without imposing language restrictions. Non-original research articles (non-peer-reviewed articles/preprints, posters, conference abstracts, protocols, book sections, book reviews, dissertations, letters to the editor, opinions, or commentaries) were excluded. All co-authors and a health science librarian developed the search strategy using keywords related to self-care and travel (S3 Appendix). Searches were initially performed on 5 October 2022 without limiting publication date and language, and updated on 30 October 2024 to capture records since inception.

### Study selection and quality assessment

A literature search was conducted, followed by manual and automated deduplication using Covidence [22]. Twenty records were selected for initial pilot screening. Title and abstract screening of the remaining articles was conducted afterwards, followed by full-text screening. Discrepancies were discussed until a consensus was reached. All processes that indicate a change in the number of articles were noted on the PRISMA flow diagram [18]. The following data were extracted from each article and tabulated: first author, publication year, aim, population, geographical location, sample size, data collection methods, and data analysis methods.

Eligible articles were assessed for quality using the CEBM Critical Appraisal of Qualitative Studies sheet, which consists of eight questions with "Yes", "No", or "Unclear" response options [23]. The questions included the study rationale,

the appropriateness of a qualitative approach, the sampling strategy, data collection methods, data analysis, the researcher's position, whether the results made sense, a result-justified conclusion, and transferability of findings. The CEBM checklist does not specify exclusion thresholds; therefore, we adopted a conservative criterion, excluding studies with more than two 'No' responses to maintain methodological rigour. This pragmatic threshold was chosen because multiple negative responses indicate significant limitations in design or reporting. Each study was independently assessed, and discrepancies were resolved through consensus. Inter-rater reliability was supported by calibration exercises and documented decision logs.

## Data synthesis

Meta-synthesis is a generic term for synthesising qualitative studies such as meta-ethnography, meta-aggregative review, meta-study, or thematic synthesis [17,24]. Quantitative findings from multi-methods and mixed-methods articles were excluded. We applied the thematic synthesis approach by Thomas and Harden [25] as it provides a clear step-by-step guide and allows for the integration of qualitative findings across diverse contexts and methodologies while preserving interpretive depth. The approach uses three stages for synthesising the data: line-by-line coding, grouping codes into one or more descriptive themes, and generating analytical themes. Coding was conducted line-by-line for all texts under the original study's 'results' or 'findings' section [25]. For multi-methods and mixed-methods articles, the texts under the section that indicates the presentation of qualitative findings were subject to extraction. Codes were grouped into descriptive themes through an iterative process of comparison and refinement. Analytical themes were generated by interpreting relationships among descriptive themes in light of the review question and theoretical frameworks.

Coding reliability was ensured through a multi-step process. Each line of the text data from each article was coded according to its meaning and content, and descriptive themes were developed by inductively grouping codes into a hierarchical tree structure based on similarities and differences. A preliminary codebook was developed during pilot coding and refined iteratively as new codes emerged. Analytical themes were generated using descriptive themes to answer the review question. Discrepancies were discussed in detail to reach a consensus, considering the meaning behind each theme and its relevance to the research question. The descriptive and analytical themes were tabulated along with at least two supporting quotes from the original studies to enhance transparency. NVivo 13 was used for data management and theme development [26]. Formal inter-coder reliability statistics were not calculated, as thematic synthesis prioritises consensus-building over quantitative agreement. All coding decisions were reviewed collaboratively to ensure consistency.

During the synthesis process, the SCM served as a sensitising concept (background ideas that shape and influence how the research problem is approached) during coding and interpretation [12,27], ensuring theoretical integration beyond the mapping. The SCM constructed self-care in four dimensions using 32 existing theories, frameworks, and models [12]. Each dimension underscores different aspects of self-care, from personal actions to broader systemic and policy influences. The Self-Care Activities dimension focuses on individual activities and capabilities for self-care and includes the Seven Pillars of Self-Care [12,28]. The Self-Care Behaviour dimension addresses actions that support positive self-care behaviours and lifestyles. The Self-Care Context considers reliance on resources and their utilisation. The Self-Care Environment focuses on drivers and barriers to self-care within the broader fiscal and policy environment, along with cultural and social factors. Codes and descriptive themes were iteratively compared against the SCM dimensions during analysis.

## Mapping to the self-care matrix

To map the meta-synthesis results with current theory in self-care, the SCM was employed [12]. A mapping procedure was developed to assess conceptual similarities between the meta-synthesis descriptive themes and the SCM's dimensions and domains. The relationships were categorised into convergence, partial convergence, and lack of convergence. A relationship was convergent when one or more descriptive themes were similar to a dimension or domain. Partial convergence occurred when a descriptive theme related to more than a single dimension or domain. A lack of convergence

occurred when a descriptive theme was unrelated to the SCM dimensions or domains. Discussions were made to address disagreements with consensus-building.

### Reflexive statement

The research team comprised health professionals from different backgrounds, including pharmacy, medicine, nursing, and public health. All members were experienced in conducting qualitative meta-synthesis (collective experience = 23 years), qualitative or mixed-methods studies (collective experience = 80 years), and research in topics of self-care (collective experience = 73 years) and travel health (collective experience = 10 years). Most team members were well-travelled globally.

## Results

### Study characteristics

Of the 2,394 articles retrieved after deduplication, 105 were assessed for eligibility (Fig 1), and 17 were included in the synthesis (Table 1).

Two studies were conducted among tourists [33,43], and 15 were among the migrants [29–32,34–42,44,45] (Table 1). The articles reported self-care practices for skin diseases [29], dengue fever [43], malaria [39,45], unspecified fever [31], type-2 diabetes [36,40–42], and cardiovascular disease [33]. Five articles did not report specific health conditions [32,34,35,37,44]. Unless otherwise specified, this section's term 'travellers' encompasses both international travellers and migrants. When assessed with the CEBM checklist, common risks for quality were the reporting of the researcher's positionality (S4 Appendix). However, all identified studies met the quality threshold for meta-synthesis and were included for analysis.

### Descriptive themes, analytical themes, and mapping to the Self-Care Matrix

We generated 21 descriptive themes (S5 Appendix) using an inductively developed codebook (S6 Appendix), considering travellers' health statuses and strategies to maintain health conditions and cope with acute or chronic illness, such as relying on self-treatment and accessing health services at and outside the destination country. The sample of quotations from primary studies for theme generations was also presented in S5 Appendix. Descriptive themes varied from individual factors, such as hygiene practices, to broader social and environmental factors, such as stigma, legal status, and discrimination. We synthesised five analytical themes to answer the review question: self-care empowerment, mutual understanding of cultures, healthcare challenges and opportunities, preventive self-care, and facilitated self-care.

All descriptive themes, except social engagement, mapped to the SCM (Fig 2). Partial convergences were observed on three descriptive themes: self-care products and medicines, self-treatment, and adherence to treatment, which were linked to Self-Care Activities (Rational use of products and services), Self-Care Context, and Self-Care Environment dimensions of the SCM. Social Engagement did not converge with the Seven Pillars of Self-Care, which are part of the SCM's Self-Care Activities dimension. This divergence is because it represents a relational determinant with social networks, i.e., social capital as a meta-contextual factor, rather than a discrete self-care activity.

### Self-care empowerment

Voluntary international travellers tended to have some degree of preparation driven by knowledge and health literacy, but could miss some important matters. Travellers prioritised personal medicines and devices on the checklist, but attending pre-travel consultations appeared lacking [30,33]. Health insurance was essential, but only for those who could access and afford it [33,34,43]. Some travellers understood possible risks related to individual health and environmental situations, but sometimes did not understand the details [33,38,45]. Most often, travellers did not know the destination

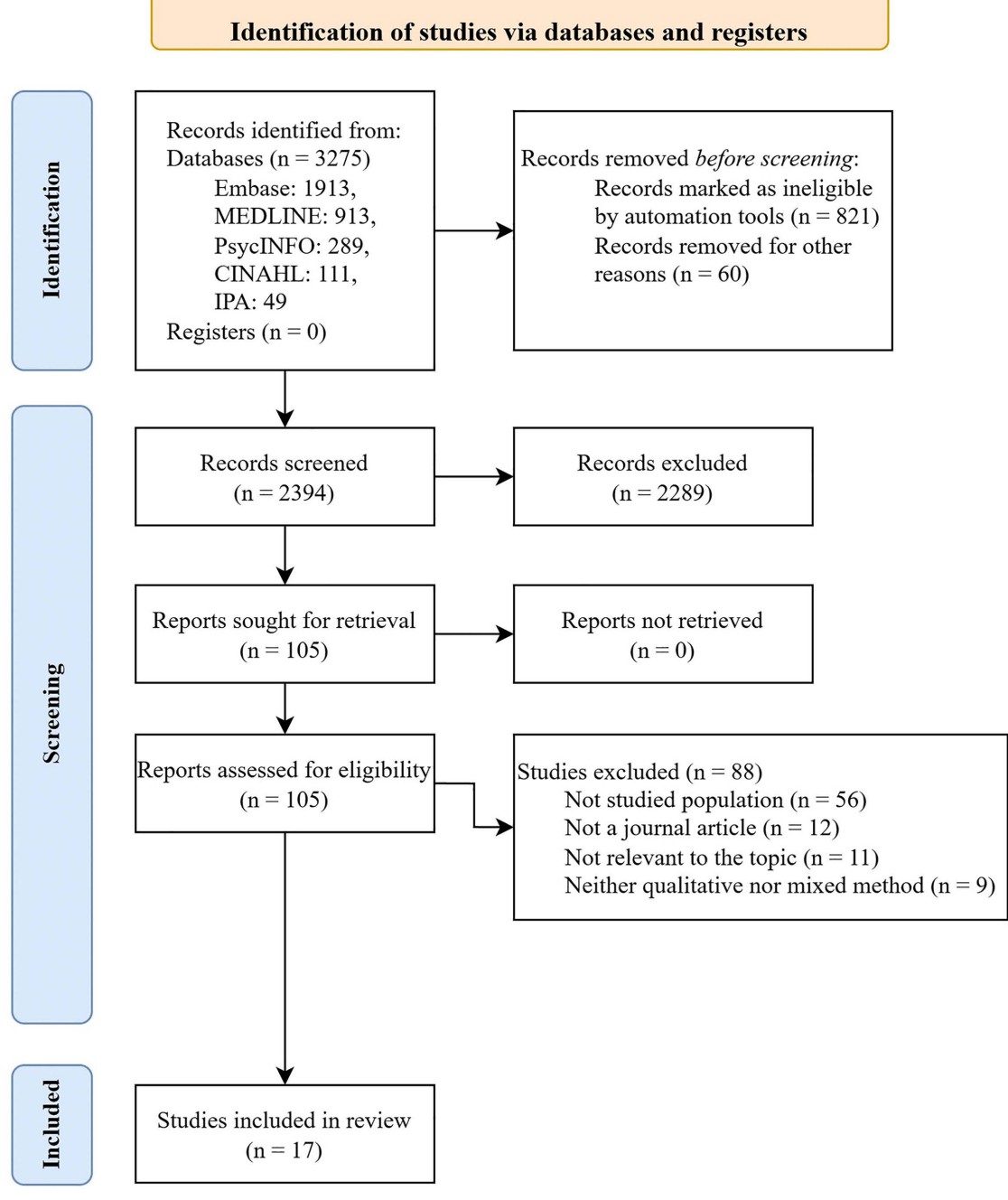

**Fig 1. PRISMA flow diagram.**

country's healthcare system until they needed to access it [30,33–35,43]. Unfortunately, navigating the health system was not always easy, especially when social capital was deficient and other barriers were present [30,33–35].

Travellers needed self-care empowerment through community interventions and initiatives to overcome those gaps. Interventions and initiatives improved travellers' knowledge, motivating them towards positive behaviour change and

**Table 1. Study characteristics.**

| No | Author(s) (Year) | Aim(s) | Study location | Sample | Data collection method | Data analysis* |
|---|---|---|---|---|---|---|
| 1 | Arcury et al. (2006) [29] | To describe the self-management of skin disease in Latino migrant and seasonal farm-workers in North Carolina | US | Latino migrant and seasonal farmworkers (N = 30, F = 6, M = 24, age range: 18–59 years, LoS unreported) | In-depth interviews | Systematic analysis |
| 2 | Fauk et al. (2022) [30] | To understand barriers to accessing HIV care services in destination countries among Indonesian, male, former (returned) migrant workers living with HIV | Indonesia | Returned Indonesian migrant workers from Malaysia, Thailand, Taiwan, and Hongkong (China) (N = 22, M = 22, age range: 25–48 years, LoS unreported) | In-depth, in-person interviews | Qualitative data analysis framework |
| 3 | Khirikoek-kong et al. (2023) [31] | To understand the concept of fever and its believed causes amongst migrants alongside the Thai-Myanmar border, and explore the association between migrants' determinants and health-seeking behaviours | Thai-Myanmar border | Migrants/community members, Village Health Volunteers, primary care staff, key informants, Tak-Province Community Advisory Board, Primary care unit staff (N = 56, gender, age, LoS unreported) | In-depth interviews, focus group discussions, participatory visual method workshop | Thematic analysis |
| 4 | Kilanowski et al. (2010) [32] | To describe the meaning of food in the family life of migrant farmworker mothers; specifically to describe their understanding of the relationship of dietary intake to health, the environmental contributors to their families' dietary intake, and the use of foods in the commemoration of family occasions. | US | Migrant farmworker mothers (N = 57, F = 57, age range: 18 ->48 years, LoS unreported) | Interviews | Thematic analysis |
| 5 | Liew et al. (2020) [33] | (1) To identify perceived barriers to travel (2) To explore potential travel health benefits (3) To generate recommendations for CVD travellers and the clinicians caring for and advising them | Ireland | Adults older than 18 years with various cardiovascular conditions travelled overseas (N = 12, F = 4, M = 8, age range: 58–82 years, LoS unreported) | In-depth interviews | Thematic analysis |
| 6 | Lin et al. (2016) [34] | (1) To examine strategies that African migrants in Guangzhou have adopted in response to healthcare barriers (2) To explore their perceptions of how to address their healthcare needs as their numbers continue to grow | China | African migrants, including businessmen, students, housewives, and English teachers (N = 35, F = 10, M = 25, age mean: 33.7 years, LoS mean: 4.4 years) | Semi-structured interviews | Unspecified |
| 7 | Madden et al. (2017) [35] | To understand the health needs and health service experiences of the Eastern European population in a town in Northern England | UK | Eastern European migrants (N = 42, F = 26, M = 16, age range: 16 – 60s years, LoS unreported) | Focus groups, one-to-one interviews, and small group interviews | Framework analysis |
| 8 | McElfish et al. (2016) [36] | To identify barriers at the organisational, community, and policy levels that constrain efforts to achieve diabetes self-management | US | Marshallese migrants (N = 69, gender, age, and LoS unreported) | Focus groups | Unspecified |
| 9 | McVea (1997) [37] | To describe the practice of self injection of antibiotics, vitamins and other medicinal compounds among migrant farmworkers in order to determine the magnitude of the public health concern for its potential to spread the HIV virus. | US | Migrant farmworkers (N = 31, F = 24, M = 7, age range: 18–53 years, LoS unreported) | Semi-structured interviews | Unspecified |
| 10 | Obach et al. (2024) [38] | To identify the barriers and facilitators that young migrants experience to access sexual and reproductive healthcare in the Tarapacá region of Chile | Chile | International young migrants from Colombia, Venezuela, and Ecuador (N = 25, F = 12, M = 13, age range: 18–29 years, LoS unreported) and health workers (N = 10) | Semi-structured interviews | Thematic analysis |

*(Continued)*

**Table 1.** (Continued)

| No | Author(s) (Year) | Aim(s) | Study location | Sample | Data collection method | Data analysis* |
|---|---|---|---|---|---|---|
| 11 | Parent et al. (2022) [39] | (1) To determine the contextual elements influencing the use of Malakit<br>(2) To understand the way gold miners perceive Malakit<br>(3) To identify the elements that are favorable and unfavorable to the use of Malakit<br>(4) To identify what can be improved in the project | French Guiana (main site), Brazil, Suriname | Gold miners working illegally (main target) (N = 20, gender, age, and LoS unreported), local actors (N = 6, gender, age, and LoS unreported), health facilitators (N = 6, gender, age, and LoS unreported) | On-site observation, semi-structured interviews, group interviews | Thematic analysis |
| 12 | Porqueddu (2017) [40] | To understand Indian and Pakistani migrants' understandings of diabetes, their experiences of the illness and their strategies for managing the disease | UK | Indian and Pakistani migrants (N = 21, gender, age, and LoS unreported) | Participant observation, group discussions, semi-structured interviews | Grounded theory |
| 13 | Shahab et al. (2019) [41] | To understand how beliefs, culture, and life experiences may contribute to the community burden of diabetes and may affect individual diabetes care | Australia | Samoan migrants (N = 20, F = 7, M = 13, age range: 36–67 years, LoS unreported) | Semi-structured interviews, field notes | Thematic analysis using a constructivist-grounded theory |
| 14 | Tyson et al. (2019) [42] | To explore the cultural and political economic factors that influence diabetes management among Hispanics residing in a rural farm working community in Florida | US | Latino farmworkers (N = 30, F = 15, M = 15, age range: 28–74 years, LoS unreported) | Semi-structured interviews | Thematic analysis |
| 15 | Vajta et al. (2015) [43] | To understand backpackers' knowledge of dengue fever and the decision-making process they use when considering utilising the Australian healthcare system | Australia | Backpackers (N = 34, gender unreported, age range: 18–30 years, LoS unreported), hostel receptionists (N = 5, gender, age, and LoS unreported), travel agents (N = 5, gender, age, and LoS unreported), pharmacists (N = 5, gender, age, and LoS unreported), government employee (N = 1, gender, age, and LoS unreported) | Semi-structured interviews, field notes | Open coding, intermediate coding |
| 16 | Westerling et al. (2020) [44] | (1) To explore the variation in implemented policies related to rational antibiotic use that citizens in Turkey and Turkish migrants in Germany, the Netherlands and Sweden are subjected to<br>(2) To discuss the implications for the promotion of rational antibiotic use | Turkey, Germany, Netherlands, Sweden | Citizens of Turkey (N = 37, F = 25, M = 12, age range: 21–50 years); Turkish migrants in Germany, the Netherlands, and Sweden (N = 47, F = 28, M = 19, age range: 23–70 years, LoS unreported); family physicians and pharmacists in those four countries (N = 45, F = 27, M = 18, age range: 26–57 years) | Focus groups, in-depth interviews | Inductive content analysis |
| 17 | Yan et al. (2020) [45] | To understand behavioural barriers and opportunities to improve testing and treatment for malaria as part of the first stage of the larger Breakthrough ACTION Guyana initiative, a social and behaviour change (SBC) project led by Johns Hopkins Center for Communication Programs (CCP) | Guyana | Gold miners (N = 70, gender, age, and LoS unreported), other mining camp staff (N = 17, gender, age, and LoS unreported), and other key stakeholders (N = 22, gender, age, and LoS unreported) | Focus groups, in-depth interviews | Framework analysis |

F: Female, M: Male, LoS: Length of Stay *Unspecified: The original authors did not explicitly specify or mention their analytical approach.

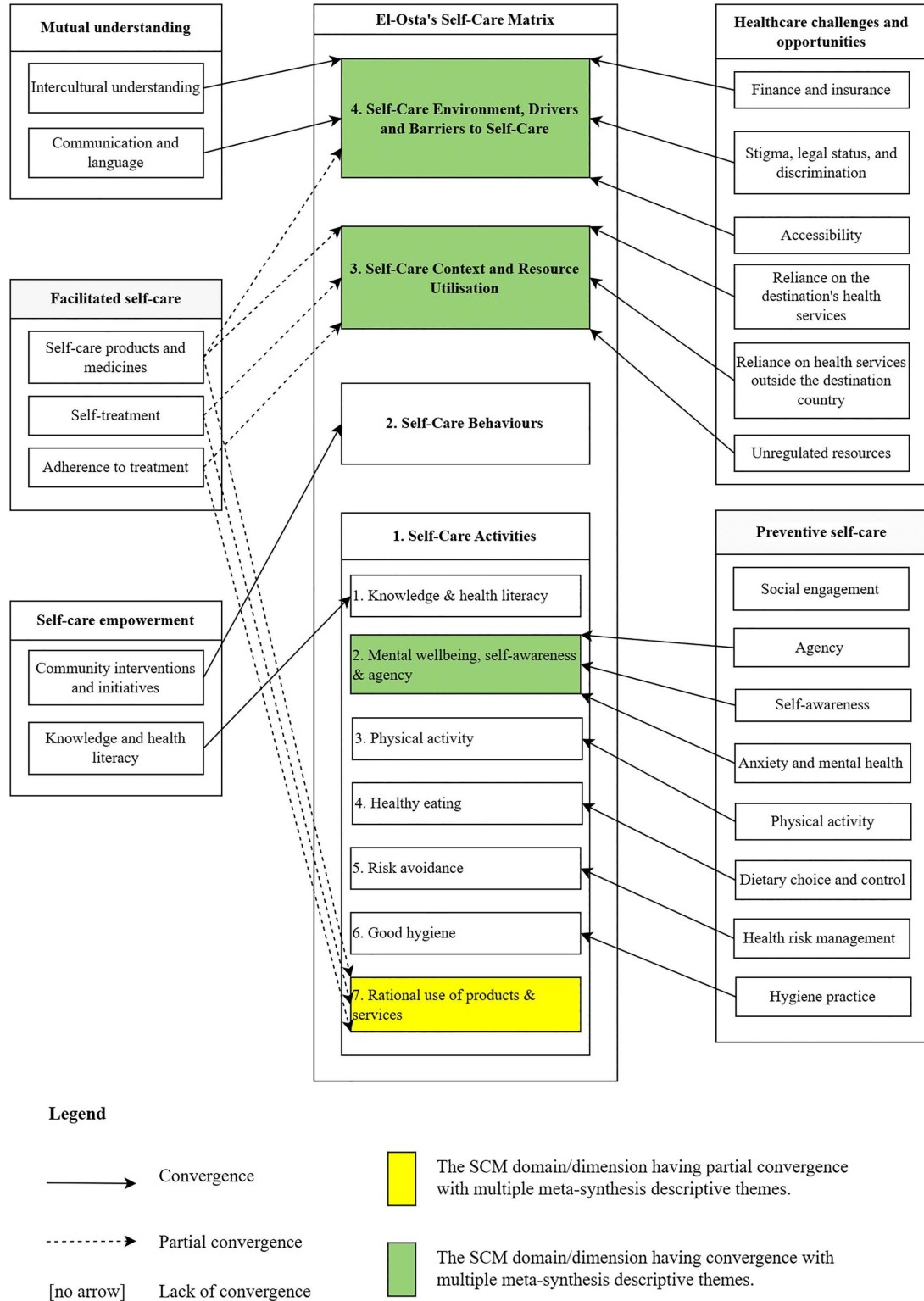

**Fig 2. Meta-synthesis descriptive themes mapped to the Self-Care Matrix.**

reducing layered barriers to healthcare access [38,39,41]. The involvement of community-based organisations and community figures, such as religious leaders, could improve the delivery and acceptance of interventions [38,41].

## Mutual understanding of cultures

As foreigners in an unfamiliar environment, travellers felt at a disadvantage due to cultural differences. Language barriers were common, as well as major cultural barriers [29–31,33,34,36]. For example, travellers with limited language struggled to explain their problems to a health professional, which could deter them from future access [30,33]. Unpleasant feelings were reported when interacting with the locals, even when an interpreter was present [33,34]. Moreover, travellers may bring cultural practices and beliefs that are at odds with the destination country's culture [30,35–37,40].

Many travellers understand that the benefits of travel include broadening their language and cultural horizons [33]. Some travellers were able to learn and adjust to the destination country's culture [35]. Meanwhile, others wanted the destination country's recognition of their cultural background, practices, and beliefs [34,36,40]. For instance, travellers suggested the integration of their languages into the destination country's health services from translated information materials [36] to the provision of health professionals who can speak their languages [34]. Therefore, there was a need for mutual understanding of cultures, i.e., travellers wanted their cultures to be understood and accommodated, and they needed to understand the destination country's culture in relation to self-care.

## Healthcare challenges and opportunities

Travellers did not typically require daily access to the destination country's healthcare services. However, when access was required, availability and accessibility were sometimes limited [30,33]. Despite language barriers, travellers often faced other challenges generated inside and outside the destination's healthcare systems, such as stigma and discrimination, legal status, and financial and insurance constraints [29,31–34,36,38,41–43].

Travellers had different solutions to overcome those challenges, such as visiting other countries [34] or returning to their country of origin [29,34,44] to obtain care from more accessible health services. Some travellers rely on themselves, family, or other immediate circles, if any, which could delay seeking care [29,31,39,43,45]. Other travellers opted for unregulated resources, such as online marketplaces [44], the black market [29], or lay injectionists [31,37]. Therefore, there is a need to reduce travellers' challenges in accessing formal healthcare to improve health outcomes.

## Preventive self-care

Practising self-care to prevent unwanted health events or deterioration of pre-existing health conditions can maximise travel benefits. Preventative self-care practices reported by travellers included maintaining hygiene to prevent skin problems and preparing food [29,32], identifying foods and beverages they thought were healthy from the unhealthy, including traditional foods [32,33,41,42], and travelling with a partner or group for help in times of health need [29,33,43]. Solo travellers sought some degree of social engagement with the locals and fellow travellers, especially those from similar countries of origin [30,33–35,37,41,43].

Not all travellers were able to practice self-care without difficulty. Travellers reported psychological challenges such as anxiety, fear, and worry that occurred as a result of both personal and environmental deficits [30,31,33,38,39,42]. For example, travellers without legal status feared being detained and deported [30,31]. Meanwhile, elderly travellers lacking physical fitness were worried about demanding physical activity, although some benefitted from increased physical activity to maintain or improve their health and fitness [33]. Other travellers struggled to manage physical activity due to unpredictable schedules and competing priorities for self-caring for their chronic illness [33,42]. Addressing these psychological concerns with preventive self-care was needed to maintain mental well-being.

## Facilitated self-care

Self-treatment in any form, such as self-medication and self-management, was often a response to layered barriers to accessing healthcare services [29–31,34,37,40,42,43,45]. Travellers used any products and services they could afford and access, even if they posed an additional risk and were not in line with the destination country's regulations and policies [29–31,34,37]. Self-treatment with herbal and traditional medicines was not always perceived as superior to treatments from an authorised health professional, with some using it for a complementary purpose [30,40].

Adherence to prescription treatment may be compromised for those on treatment for acute or chronic conditions [30,33,40,45]. Some travellers intended not to adhere to the treatment they received due to side effects, for example [40,45]. Others became non-adherent due to uncontrollable factors, such as the mobile nature of travel and inaccessibility to products and services [30,33,40,45].

Depending on their perceived necessity and severity of an illness, travellers went to the nearest accessible health facility, with those offering free services preferred, such as community pharmacies, which provided free consultations [29,39,43,45]. Facilitation by a health professional improved travellers' adherence to treatment [45]. Therefore, there was a need for facilitated self-care from health professionals and facilities that are accessible to travellers.

## Discussion

To our knowledge, this is the first review to explore self-care needs among international travellers using thematic synthesis, addressing a knowledge gap in unfamiliar environments. This review primarily synthesises evidence from transient migrants, with limited data on short-term travellers. Findings should be interpreted with this scope in mind. Overall, our findings highlight the importance of social engagement and the need to recognise the social aspect of self-care. Our study advances theory by the integration of social capital into the SCM, addressing a gap in travel and migration health literature.

The SCM [12] was a suitable framework by which we could conduct our analysis. However, we identified two key issues for consideration in the SCM as a sensitising concept in our analysis. First, we identified overlapping and interconnected terms, such as 'knowledge' and 'health literacy', and 'mental wellbeing', 'self-awareness', and 'agency', which were combined to form an SCM domain or theme. While the use of overlapping concepts offers advantages, such as providing a holistic and realistic understanding of human experience, it can lead to conceptual ambiguity, making it difficult to define, distinguish, and measure constructs accurately. This issue may stem from the development process of the SCM, which brought together 32 self-care models, frameworks, concepts, and theories. Accordingly, we suggest refining the framework through expert consensus. Second, our study's findings demonstrate the value of social interaction for self-care. Specifically, social engagement and social capital can enrich the SCM's Self-Care Activity and Self-Care Context dimensions, respectively.

We propose adding social engagement as a self-care component for international travel. People have social capital to support health outcomes in a familiar environment. Social capital can be understood as the value of the individuals' or groups' social networks [7,8]. Social capital may decrease when moving to a new, unfamiliar environment. Social engagement can build social capital. Satisfaction with social networks was correlated with satisfaction with social support, leading to a sense of belonging to the community and overall happiness in the destination country [8]. Social capital increases among international students engaged in campus organisations due to the more extensive social networks they develop [46]. In addition, social engagement can positively impact social determinants of health. For example, international students receiving group training in social skills and adaptation to self-help groups improved their social capital and mental health [47]. Findings from this meta-synthesis have identified that social engagement is a form of social capital important for travellers' self-care.

Among travellers, social capital can be developed in the destination countries by engaging with others, such as fellow travellers, existing traveller communities, or the locals. Travellers gain different benefits from whom they socially engage

because self-care is a social construct [48] and an individual's social environment shapes their self-care knowledge, attitudes, and practices. For example, engaging with fellow travellers helps them practise preventive self-care, such as increased physical activity. Engaging with existing traveller communities or the locals helps travellers access and navigate the destination country's healthcare system, overcome language barriers, and become accustomed to the destination country's culture and customs around health maintenance. Existing traveller communities, especially those with similar cultural or national backgrounds, help travellers have a sense of belonging and maintain cultural self-care practices such as self-treatment with herbal and traditional medicines. This meta-synthesis found that what travellers used to know and practice in their country of origin will be brought along when they travel to other countries, despite possible incompatibility with mainstream systems and practices at the destination countries, for example, self-injection.

Our findings extend current discourse on global migration health by highlighting how mobility intersects with self-care practices in ways that are increasingly mediated by digital technologies. Emerging evidence suggests that mobile health applications, remote consultations, and digital health literacy interventions can serve as critical enablers of self-care for migrants and travellers, mitigating barriers to continuity of care and access to health information [49,50]. This aligns with WHO's self-care agenda, which positions digital tools as central to empowering individuals [51]. By situating our results within this ground, we contribute novel insights into how self-care strategies adapt under conditions of global mobility, offering implications for inclusive digital health design and equity-focused policy development.

Seemingly incompatible self-care practices with the destination country's policies and practices can be addressed through several findings from this study, i.e., self-care empowerment, reducing healthcare challenges, and self-care facilitation. An example of self-care empowerment can be supplying accurate and appropriate information pertaining to travel health, including possible health risks, through accessible outlets. Inappropriate information potentially promotes harmful consequences rather than self-care [16]. Pre-travel health consultation in the country of origin is one outlet for seeking appropriate health information [52]. However, considering the low utilisation of this pre-travel service primarily due to the perceived low risk of infections [52], the destination country should tailor interventions and initiatives that provide travel-health information to target travellers. In fact, there is a demand for strategically disseminating health information and promotional messages to help travellers practise self-care [53].

Regarding healthcare challenges, most sick travellers delay seeking care due to multi-layered access barriers. The healthcare access barriers identified in this study have confirmed previous studies [10,11]. Our findings also confirm past studies that inadequate access leads to unregulated sources, products, and services, especially driven by more affordable costs [54]. Efforts to reduce access barriers, such as providing translation tools or interpreters and improving accessibility and affordability, should be selected carefully to prevent harmful practices. Reliance on unregulated medicine sources due to challenging access to healthcare raises ethical and public health concerns, including antimicrobial resistance [55]. Addressing these risks requires targeted health education and regulatory interventions.

Regarding self-care facilitation, sick travellers who are delayed seeking care from clinics or hospitals may seek care from a community pharmacy. Pharmacies are more numerous and widespread than referral facilities, offering better community accessibility. Previous research explored the interactions between community pharmacies and foreign-speaking customers [56,57]. Participants wanted medication information, including dosage, side effects, administration techniques, and missed doses from the pharmacies [56]. These interactions allow pharmacists to offer self-care facilitation services, assisting travellers in obtaining effective self-treatment and improved adherence for those already in therapy.

### Limitations

There were five limitations in our study, including the migration status, researcher's positionality, and research focus of the original studies, the COVID-19 context, and the risks of publication and selection bias. Firstly, establishing the travel or migration status of participants was challenging. However, this challenge was related to the body of literature, where some studies did not clearly define the participants' migration status. Despite the small number of articles included in this review,

the included studies reported data from a total of 769 participants, with two studies having more than 100 participants due to the multi-site settings and multiple stakeholders involved. Secondly, the majority of included studies (12 out of 17) did not report the researcher's positionality, preventing us from conducting sensitivity analysis by excluding them to reassess the final themes. The qualitative design of the included studies also prevented us from conducting a weighting analysis. Thirdly, most of the synthesised studies focused on either an illness or a phenomenon, making this review potentially missing the full range of health problems faced by travellers and migrants. However, our search strategies did not limit the year of publication, ensuring a comprehensive retrieval of the current knowledge of the topic. Fourthly, none of the studies included in this meta-synthesis focused on the COVID-19 pandemic, which may impact the magnitude of self-care practices among travellers and migrants. However, this limitation also showed the scarcity of studies on travellers and migrants during the COVID-19 pandemic. Fifthly, excluding grey literature may introduce publication bias, as unpublished qualitative studies could provide additional perspectives. Also, including only qualitative components in mixed-method studies may introduce selection bias. To address potential publication and selection bias, we conducted a sensitivity check by excluding three borderline studies with two 'No' responses in the quality assessment. The five analytical themes remained unchanged, with only one descriptive theme (i.e., 'Health Risk Management') disappearing, suggesting that the synthesis was sufficiently robust to this exclusion.

## Conclusion

This study has identified five primary self-care needs: self-care empowerment, mutual understanding of cultures, health-care challenges and opportunities, preventive self-care, and facilitated self-care. Our study reveals that when people travel or migrate, they frequently lose social connections that contribute to their overall well-being. Strengthening established social ties and the development of new social ties are key to effective self-care. This review contributes to self-care theory in the context of international mobility by advancing social capital as a meta-contextual factor and offers practical guidance for policy and digital health interventions.

## Supporting information

**S1 Appendix. ENTREQ statement.**
(DOCX)

**S2 Appendix. PRISMA 2020 checklist.**
(DOCX)

**S3 Appendix. Search strategies.**
(DOCX)

**S4 Appendix. Quality assessment of studies.**
(DOCX)

**S5 Appendix. Analytical and descriptive themes with quote examples.**
(DOCX)

**S6 Appendix. Codebook examples and thematic tree.**
(DOCX)

## Acknowledgments

ANWP acknowledges the support from the Indonesian Endowment Fund for Education (LPDP) Scholarship to study a PhD at The University of Sydney School of Pharmacy, New South Wales, Australia.

## Author contributions

**Conceptualization:** Antonius Nugraha Widhi Pratama, Brahmaputra Marjadi, Rebekah Jane Moles, Carl Richard Schneider.

**Data curation:** Antonius Nugraha Widhi Pratama.

**Formal analysis:** Antonius Nugraha Widhi Pratama, Jack Charles Collins, Carl Richard Schneider.

**Funding acquisition:** Antonius Nugraha Widhi Pratama.

**Investigation:** Antonius Nugraha Widhi Pratama.

**Methodology:** Antonius Nugraha Widhi Pratama, Brahmaputra Marjadi, Rebekah Jane Moles, Carl Richard Schneider.

**Project administration:** Antonius Nugraha Widhi Pratama.

**Supervision:** Brahmaputra Marjadi, Rebekah Jane Moles, Carl Richard Schneider.

**Writing – original draft:** Antonius Nugraha Widhi Pratama.

**Writing – review & editing:** Antonius Nugraha Widhi Pratama, Brahmaputra Marjadi, Jack Charles Collins, Rebekah Jane Moles, Carl Richard Schneider.

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
