## [Decision Letter · Decision Letter 0]

30 Oct 2025

Dear Dr. Pratama,

Thank you for submitting your manuscript to PLOS ONE. After careful consideration, we feel that it has merit but does not fully meet PLOS ONE’s publication criteria as it currently stands. Therefore, we invite you to submit a revised version of the manuscript that addresses the points raised during the review process.

We look forward to receiving your revised manuscript.

Kind regards,

Wejdan Shahin

Academic Editor

PLOS ONE

Journal Requirements:

2. We note that your Data Availability Statement is currently as follows: All relevant data are within the manuscript and in Supporting Information files.

Additional Editor Comments:

Overall, the study addresses a valuable topic but needs clearer framing and stronger methodological justification. You need to thoroughly revise the reviewers' comments to enhance the quality of the manuscript. Besides reviewers' comments please clarify the distinction between key concepts such as self-care, and self-management.

The quality appraisal needs to mention all the criteria that were assessed.

The discussion needs better interpretations and not just repeating of the findings of the included studies.

Reviewers' comments:

Reviewer's Responses to Questions

**Comments to the Author**

1. Is the manuscript technically sound, and do the data support the conclusions?

Reviewer #1: Partly

Reviewer #2: Partly

Reviewer #3: Partly

Reviewer #4: Partly

2. Has the statistical analysis been performed appropriately and rigorously?

Reviewer #1: N/A

Reviewer #2: N/A

Reviewer #3: Yes

Reviewer #4: No

3. Have the authors made all data underlying the findings in their manuscript fully available?

Reviewer #1: Yes

Reviewer #2: Yes

Reviewer #3: Yes

Reviewer #4: No

4. Is the manuscript presented in an intelligible fashion and written in standard English?

Reviewer #1: Yes

Reviewer #2: Yes

Reviewer #3: Yes

Reviewer #4: Yes

Reviewer #1: Dear Authors,

Thank you for giving the opportunity to read on the self-care needs of international travellers and migrants through a systematic review and meta-synthesis of qualitative studies. I believe its quality can be improved with some considerations and incorporations as listed below.

The manuscript introduces self-care matrix (SCM) as the mapping framework, but the theoretical integration is only partial. Please clarify how SCM guided your analysis rather than being applied only at the end for thematic alignment. The concepts of ‘self-care’ and ‘self-care needs’ are used interchangeably with ‘self-management’, ‘self-efficacy’, and ‘empowerment’. A stronger conceptual delineation is necessary in the Introduction and Discussion sections to avoid ambiguity. Authors should consider expanding the theoretical background to discuss how migration, transience, and social capital interact to shape self-care practices.

The Methods section should explicitly explain the distinct populations of migrants and international travelers in health research. The inclusion and exclusion criteria should explicitly state how temporary migrants (e.g., seasonal workers) differ from permanent migrants in your dataset and how this distinction was managed during synthesis. The search strategy in Table 1 is excessively detailed for the main text. You can move it to supplementary materials to improve readability. The quality assessment process using the CEBM checklist is mentioned but not fully justified. Clarify why studies with ‘two or more ‘No’ responses’ were excluded and indicate how inter-rater reliability was maintained during screening and coding.

The analytical themes are somewhat descriptive. Figure 2 (mapping to SCM) lacks sufficient explanation in the text. Provide examples showing why certain themes (e.g., ‘social engagement’) did not converge and what this implies for the SCM framework’s completeness. Several descriptive themes overlap (e.g., ‘health literacy’, ‘knowledge’, and ‘mutual understanding’). The discussion somewhat restates findings rather than critically analysing them in light of prior research on global mobility, health inequalities, and self-care frameworks. You can discuss the role of digital health tools, remote consultations, or mobile health applications for migrants and travellers as emerging enablers of self-care. Clarify how the study contributes new insights to WHO’s self-care agenda or global migration health discourse.

Tables (e.g., Table 4) run several pages. Consider summarizing the major analytical findings and moving extended quotations to supplementary files. Figures 1 and 2 need higher resolution and clearer legends. References require formatting consistency throughout as per the journal guidelines.

The paper at this stage requires substantial revisions in (1) strengthening conceptual clarity around ‘self-care needs’, (2) improving methodological transparency and synthesis rigor, and (3) deepening the interpretive discussion linking findings to theory and practice. Addressing these issues will significantly strengthen the study’s credibility and relevance. I recommend the author to revise the manuscript with the above concerns before making a resubmission to this journal. I hope you will consider and incorporate the comments to improve the quality of the work and further make it publication ready. Best wishes.

Reviewer #2: Thank you for your efforts in trying to add knowledge to this important field of study. However, there are some pertinent concerns that have to be addressed.

General comments:

1) The objective and research question of the review are not clearly stated that the synthesis will answer.

2) This manuscript is co-authored by many and in the methods section, the audience is interested to know what research method was used and the reasons for choosing that method, the participants and sample size, how they were included or excluded from the study, setting, data collection and analysis methods used, limitations and strengths of the study and the IRB authorization if required. There is a section reserved for the participation of each author, which is not in the methods section. Please revise this section to reflect single team dynamics.

3) This is a meta synthesis literature review. Could you please explain the interpretive and iterative process used in the study to identify themes?

4) How did you assess the quality of the included studies? Were there predetermined criteria?

5) In a layman's term, what is the novel knowledge or insights generated by this study?

Reviewer #3: The study focuses on the self-care needs of “traveller” and “migrant” groups with short- or medium-term international mobility, using a systematic review and thematic meta-synthesis of qualitative studies. Searches were conducted in MEDLINE, Embase, IPA, PsycINFO, and CINAHL without date or language restrictions, resulting in the inclusion of 17 studies (n=769). The authors followed ENTREQ and PRISMA reporting standards, used the CEBM qualitative checklist for quality appraisal, applied the three-stage Thomas & Harden synthesis approach, and mapped findings onto the Self-Care Matrix (SCM), identifying five analytical themes. The theme of “social participation” not fully aligning with SCM’s “Seven Pillars” constitutes an important theoretical finding. Overall, the topic is timely, the methodological framework is appropriate, and the findings are meaningful for policy and practice; however, the sample’s overwhelming focus on “migrants,” the conceptual distinction between “traveller” and “migrant,” and certain methodological details require clarification.

Only two of the 17 studies focused directly on “travellers” (tourists or backpackers); the remaining studies concentrated on “migrants,” mostly from low-income or context-specific backgrounds. This imbalance limits the transferability of results to both populations mentioned in the title. Therefore, the title, objectives, and discussion should be reframed more transparently and equitably to reflect the actual scope.

The database selection is appropriate; the update date (30 October 2024) is provided, and a PRISMA flow diagram is reported. However, since no grey literature, projects, or reports were included, publication bias in qualitative studies should be discussed.

The inclusion and exclusion criteria are clearly described; however, the exclusion of “refugee/asylum seeker” groups requires further ethical, sociopolitical, and experiential justification. While it is understandable that forced migrants may have distinct self-care patterns, the impact of excluding them based on the “transient” criterion should be explicitly discussed.

Using the CEBM qualitative checklist and excluding studies with more than two “No” responses is appropriate. Nevertheless, a detailed summary matrix for each study (e.g., the distribution of “Yes/No/Unclear” answers per criterion) should be added as an appendix. Researcher positionality (reflexivity) was often underreported; this may require weighting studies in the synthesis according to their methodological transparency.

The use of the Thomas & Harden approach is appropriate; however, the reliability of the coding process (dual coding, consensus procedures, codebook examples, and sample coded excerpts) should be described in more detail. NVivo was mentioned, but no information was given regarding inter-coder agreement or comparative control.

Including only the qualitative parts of mixed-methods studies is methodologically correct; however, this selective inclusion introduces a potential selection bias that could be tested through a brief sensitivity analysis (e.g., checking whether themes remain stable when borderline cases are excluded).

The convergence logic of “full/partial/none” mapping to the SCM is clear, but the process should be supported by independent dual assessment and description of how disagreements were resolved. The authors should also strengthen the theoretical reasoning for why “social participation” did not align with the Seven Pillars (e.g., by framing social capital as a meta-contextual factor influencing self-care behaviors).

Within the theme “self-care facilitation,” the grouping of product/medicine access, self-treatment, and adherence behaviors is well structured; however, the safety and antibiotic resistance risks associated with unregulated sources (markets, online vendors, traditional healers) should be more critically discussed from ethical and public health perspectives.

The finding on social capital/participation is one of the study’s most original contributions. It should be more visibly linked to literature on help-seeking, network support, and language/cultural mediation (ideally illustrated through a conceptual model diagram).

ENTREQ and PRISMA adherence is noted; however, the full PRISMA checklist and all search strategies (for each database) should be presented in the supplementary materials. Tables describing study contexts and samples (age, gender distribution, migration status, duration of stay) should be standardized as much as possible.

Terminology should be operationally defined at the beginning of the article and used consistently throughout—particularly the distinctions among “traveller,” “migrant,” “transient,” and “international student.”

The PROSPERO registration is appropriate. The data availability statement (“All relevant data are within the manuscript and its supporting information files”) is acceptable, but adding a de-identified codebook, thematic tree, sample coded excerpts, and quotation list as a “Supporting File” would enhance transparency and reusability.

The PRISMA figure’s readability should be improved; exclusion reasons should be summarized categorically. For Tables 2 and 3, studies listed as “data analysis unspecified” should include footnotes explaining this uncertainty and its potential effect on weighting.

Reviewer #4: 1. The study presents the results of original research.

The authors collected and analyzed original empirical data (systematic publication review databases and survey results). The novelty of the study lies in identifying the five most relevant self-care areas (hygiene and self-care) for international travelers and migrants. However, the need to use the complex methodological apparatus chosen by the authors to identify these areas is questionable. If the originality of the study is ensured solely by an extremely cumbersome apparatus, then the result appears theoretically weak: the research costs outweigh the results and do not withstand Occam's razor. I recommend that the authors more precisely define the limitations of their study in the "Limitations" section and clearly indicate where they achieved a scientific breakthrough (novelty) and what the apparatus used did not allow them to achieve. Furthermore, the "Conclusion" section does not contain general conclusions, as if the authors conducted the study solely to learn that research should be conducted. This is a theoretical oxymoron: they studied at length, but what they learned is unclear. The conclusion needs to be seriously revised and significantly strengthened theoretically.

2. Results reported have not been published elsewhere.

The authors rely on the analysis of collected material that has not been published before.

3. Experiments, statistics, and other analyses are performed to a high technical standard and are described in sufficient detail.

The authors presented the study's algorithm and the statistics that influenced the results in sufficient detail. However, the authors' preliminary conclusions raise doubts. Overcoming subjective limitations is not an end in itself. Clear criteria for qualitative evaluation are needed. The authors need to propose to the scientific community ways to replicate their results so that the feasibility of achieving them becomes clear. Otherwise, the obtained results cannot be considered scientific.

4. Conclusions are presented in an appropriate fashion and are supported by the data.

Some preliminary conclusions appear well-founded and supported by statistics. However, the criteria for consensus among the research team members in formulating a general conclusion on the theoretical value of the analyzed scientific literature are unclear. The final conclusion requires significant revision.

5. The article is presented in an intelligible fashion and is written in standard English.

The ratio of professional and everyday vocabulary within the standards of scientific style. The structure of the article reflects the logic of scientific research. The style of link formatting should be adjusted according to the editorial requirements.

6. The research meets all applicable standards for the ethics of experimentation and research integrity.

The author does not violate standard standards of experimental ethics and research integrity.

7. The article adheres to appropriate reporting guidelines and community standards for data availability.

The author adheres to relevant reporting guidelines and standards for data accessibility accepted in the scientific community.

**Do you want your identity to be public for this peer review?** For information about this choice, including consent withdrawal, please see our Privacy Policy

Reviewer #1: No

Reviewer #2: **Yes:** Dr. Agnes Ebotabe Arrey

Reviewer #3: No

Reviewer #4: **Yes:** Gennady V. Bakumenko

---

## [Author Response · Author response to Decision Letter 1]

13 Nov 2025

We have provided a full response to the academic editor and reviewers in the Response to Editor and Reviewers file attached in this submission. Copying its content and pasting it to this box will damage the format, decreasing the readability.

---

## [Decision Letter · Decision Letter 1]

12 Dec 2025

Dear Dr. Pratama,

Thank you for submitting your manuscript to PLOS ONE. After careful consideration, we feel that it has merit but does not fully meet PLOS ONE’s publication criteria as it currently stands. Therefore, we invite you to submit a revised version of the manuscript that addresses the points raised during the review process.

We look forward to receiving your revised manuscript.

Kind regards,

Adetayo Olorunlana, Ph.D.

Academic Editor

PLOS One

Journal Requirements:

Reviewers' comments:

Reviewer's Responses to Questions

**Comments to the Author**

Reviewer #2: (No Response)

Reviewer #3: All comments have been addressed

2. Is the manuscript technically sound, and do the data support the conclusions?

Reviewer #2: Yes

Reviewer #3: Yes

3. Has the statistical analysis been performed appropriately and rigorously?

Reviewer #2: Yes

Reviewer #3: Yes

4. Have the authors made all data underlying the findings in their manuscript fully available?

Reviewer #2: Yes

Reviewer #3: Yes

5. Is the manuscript presented in an intelligible fashion and written in standard English?

Reviewer #2: Yes

Reviewer #3: Yes

Reviewer #2: The authors have not adequately addressed the comments referring to portraying the study as teamwork and not singling out parts played by individual researchers in the methods section. Readership is not interested in who did what but would like to know the methodological flow of the study. For example, mentioning that "Researcher ANWP coded each line of the text data..." is irrelevant in the methods section. Please address all similar sentences. There is a section reserved for that.

Reviewer #3: The manuscript has now been clearly revised and is suitable for publication. Most of the previous critiques have been addressed systematically: the “traveller vs. migrant” distinction has been clarified in the title, abstract, and methods section, with both groups defined as sharing a common context of temporary mobility. The potential bias introduced by excluding grey literature has been acknowledged in the “Limitations” section. The CEBM quality appraisal table has been provided in detail in the supplementary files (S4 Appendix). The coding process has been strengthened with two coders, a third reviewer, calibration exercises, and decision logs, substantially enhancing the reliability of the analysis. The SCM mapping was conducted independently by two researchers and validated through a consensus-building approach. A “Reflexive Statement” section (pp. 17–18) has been added, outlining the team’s experience. A sensitivity analysis has been included in the “Limitations” section (pp. 33–34), demonstrating that the analytical themes remained stable. The PRISMA diagram has been refined, Table 1 has been improved, and sample information is now clearer.

**Do you want your identity to be public for this peer review?** For information about this choice, including consent withdrawal, please see our Privacy Policy

Reviewer #2: **Yes:** Dr. Agnes Ebotabe Arrey

Reviewer #3: No

---

## [Author Response · Author response to Decision Letter 2]

12 Dec 2025

We attach Response to Reviewers document in this submission. Copying and pasting our responses to this box will distort the format and reduce readability.

---

## [Decision Letter · Decision Letter 2]

22 Feb 2026

Self-care needs among international migrants and travellers: A systematic review and meta-synthesis

PONE-D-25-50169R2

Dear Dr. Pratama,

We’re pleased to inform you that your manuscript has been judged scientifically suitable for publication and will be formally accepted for publication once it meets all outstanding technical requirements.

Kind regards,

Adetayo Olorunlana, Ph.D.

Academic Editor

PLOS One

Additional Editor Comments (optional):

Reviewers' comments:

Reviewer's Responses to Questions

**Comments to the Author**

Reviewer #2: All comments have been addressed

2. Is the manuscript technically sound, and do the data support the conclusions?

Reviewer #2: Yes

3. Has the statistical analysis been performed appropriately and rigorously?

Reviewer #2: N/A

4. Have the authors made all data underlying the findings in their manuscript fully available?

Reviewer #2: Yes

5. Is the manuscript presented in an intelligible fashion and written in standard English?

Reviewer #2: Yes

Reviewer #2: Thank you for adequately addressing all the comments to improve comprehension and clarity of your manuscript.

**Do you want your identity to be public for this peer review?** For information about this choice, including consent withdrawal, please see our Privacy Policy

Reviewer #2: **Yes:** Dr Agnes Ebotabe Arrey

---

## [Editor Report · Acceptance letter]

PONE-D-25-50169R2

PLOS One

Dear Dr. Pratama,

I'm pleased to inform you that your manuscript has been deemed suitable for publication in PLOS One. Congratulations! Your manuscript is now being handed over to our production team.

Kind regards,

on behalf of

Associate Professor Adetayo Olorunlana

Academic Editor

PLOS One